# Developing the Pneumonia-Optimized Ratio for Community-acquired pneumonia: An easy, inexpensive and accurate prognostic biomarker

**Vinícius Ferraz Cury**[1], **Lucas Quadros Antoniazzi**[1], **Paulo Henrique Kranz de Oliveira**[1], **Wyllians Vendramini Borelli**[2], **Sainan Voss da Cunha**[1], **Guilherme Cristianetti Frison**[1], **Enrico Emerim Moretto**[1], **Renato Seligman**[1,3]*

**1** School of Medicine, Federal University of Rio Grande do Sul, Porto Alegre, Brazil, **2** Neurology Service, Hospital de Clínicas de Porto Alegre, Porto Alegre, Brazil, **3** Internal Medicine Service, Hospital de Clínicas de Porto Alegre, Porto Alegre, Brazil

* reseligman@hcpa.edu.br

**Data Availability Statement:** All relevant data are within the manuscript and its Supporting Information files.

## Abstract

### Introduction

Community-acquired pneumonia (CAP) is still a major public health problem. Prognostic scores at admission in tertiary services may improve early identification of severity and better allocation of resources, ultimately improving survival. Herein, we aimed at evaluating prognostic biomarkers of CAP and a Pneumonia-Optimized Ratio was created to improve prognostic performance.

### Methods

In this retrospective study, all patients with suspected Community-acquired pneumonia aged 18 or older admitted to a public hospital from January 2019 to February 2020 were included in this study. Blood testing and clinical information at admission were collected, and the primary outcome was overall survival. CURB-65 scores and prognostic biomarkers were measured, namely Neutrophil-to-Lymphocyte Cell Ratio (NLCR), Platelet to Lymphocyte ratio (PLR), Monocyte to Lymphocyte Ratio (MLR). A Pneumonia-Optimized Ratio (POR) score was created by selecting the biomarker with larger accuracy (NLCR) and multiplying it by the patients' CURB-65 score. Multivariate regression model was performed and ROC curves were created for each biomarker.

### Results

Our sample consisted of 646 individuals (median 66 years [IQR, 18–103], 53.9% females) with complete blood testing at the time of admission. Patients scored 0–1 (323, 50%), 2 (187, 28.9%), or 3 or above (122, 18.9%) in the CURB-65, and 65 (10%) presented the primary outcome of death. POR exhibited the highest Area Under Curve (AUC) in the ROC analysis (AUC = 0.753), when compared with NLCR (AUC = 0.706), PLR (AUC = 0.630)

**Funding:** The authors received no specific funding for this work.

**Competing interests:** The authors have declared that no competing interests exist.

and MLR (AUC = 0.627). POR and NLCR presented increased crude mortality rate in the fourth quartile in comparison with the first quartile, and the fourth quartile of NLCR had more days of hospitalization than the first quartile (11.06±15.96 vs. 7.02±8.39, p = 0.012).

## Conclusion

The Pneumonia-Optimized Ratio in patients with CAP showed good prognostic performance of mortality at the admission of a tertiary service. The NLCR may also be used as an estimation of days of hospitalization. Prognostic biomarkers may provide important guidance to resource allocation in resource-limited settings.

## Introduction

Community-acquired Pneumonia (CAP) is a very common and prevalent disease. It presents a varying incidence that ranges from 1.5 up to 14 cases per 1000 persons-year in the world [1], with 598.668 hospital admissions and 52.776 deaths in Brazil in 2017 [2]. Despite efforts toward diagnosis and treatment, it is still a potentially fatal condition [3, 4] especially in low and middle-income countries. A cornerstone of the treatment of CAP is to establish a prognosis to evaluate the right management or even the necessity of admission to a specialized service [5]. Although there are advances in microbiological analysis, studies have shown that pathogen identification in CAP cases is possible in less than 40% of the total [6–8]. Within this context, prognostic biomarkers are increasingly used in clinical practice, and they provide guidance to correct allocation of resources in budget-limited settings. The Pneumonia Severity Index (PSI) and the CURB-65 (acronym for Confusion, Urea, Respiratory rate, Blood pressure and 65 years-old) are the most common severity prognostic tools used for CAP [9, 10].

The prognostic accuracy for mortality in CAP of several biomarkers such as CRP, PCT, ProADM, Copeptin, NLCR, neutrophil and lymphocyte count percentages has been assessed in many studies in different settings, and they may be particularly useful in resource-limited countries. And even the combination of biomarkers can increase their accuracy. Zhang HF et al. [11] investigated the efficacy of PSI combined with NLCR in predicting 30-day mortality in CAP patients. NLCR and PSI predicted mortality, but the combination of NLCR with PSI improved the accuracy and sensitivity. Ge YL et al. [12] investigated the value of CURB-65 score and NLCR in the prognosis of CAP, with outcomes defined as ICU admission and 30-day mortality. The CURB-65 and NLCR were independent predictors of unfavorable outcomes. NLCR was superior to CURB-65. In addition, the NLCR combined with CURB-65 showed better sensitivity and specificity.

The immune system of healthy patients with CAP promotes an increase of neutrophils and a decrease in total lymphocyte values [13]. Both indicate bacteremia and, thus, combining them on a single index could lead to a better evaluation of critically ill patients. This biomarker, called neutrophil-lymphocyte count ratio (NLCR), was used in the assessment of patients at intensive care units and was found to be associated with severity and mortality, sharing the same results as SOFA score and APACHE II [14]. Other studies presented the ratio as an outcome predictor in oncologic conditions like breast, ovary, and lung cancers [15]. Besides, NLCR was better linked to bacteremia in comparison to other infection markers [16], with NLCR > 7 as an independent factor for CAP mortality. Lately, it appears to be superior to inflammation-based scores as a predictor of lethality for CAP [11, 17]. Finally, community-acquired pneumonia is an important cause of hospital admissions and death, endorsing the

need for higher accuracy methods on the early evaluation of high-risk patients, as well as their adequate management. Few studies have explored the prognostic value of the NLCR, a low-cost and widely accessible marker, as blood counts are regularly asked at the admission of a patient with suspected CAP. In this study, we aimed at evaluating different prognostic blood count biomarkers and compared their accuracy for mortality in community-acquired pneumonia.

## Methods

This is a retrospective observational study that included all patients aged 18 or older admitted to the Hospital de Clínicas de Porto Alegre (HCPA)—a tertiary public hospital with 831 beds located in Porto Alegre, Brazil—with suspected CAP from January 2019 to February 2020. The CAP diagnosis was defined by the health care team using the Brazilian Guidelines for CAP in Immunocompetent Adults—2009 criteria [18]. They included symptoms of acute lower respiratory tract infection (cough and one or more of: sputum, shortness of breath and chest pain); focal findings on physical examination of the chest, and systemic manifestations (confusion, headache, sweating, chills, myalgia and temperature above 37.8˚C), which should be corroborated by the presence of a new lung opacity detected by chest X-ray. Exclusion criteria were previous or new diagnosis of HIV and active tuberculosis. In addition, when patients had more than one hospitalization during the study period, only the first one was selected for analysis. All medical records were reviewed to confirm CAP diagnosis, blood testing was collected for analysis purposes and data were extracted in order to calculate CURB-65 scores. This study was approved by the local ethics committee, and patients informed consent was waived by the ethics committee as this is a retrospective observational study. Confidentiality was fully maintained in accordance with the Declaration of Helsinki.

Data extracted from online database, collected at the patients admissions, were age, gender, ICD-10 (International Statistical Classification of Diseases and Related Health Problems) at admission and at discharge, blood count at admission (total white blood cell count, neutrophils, rods, eosinophils, basophils, monocytes, lymphocytes, plasma cells, platelets), besides prothrombin time (PT), partial thromboplastin time (PTT), CRP, lactate, fibrinogen, ferritin, bilirubin, creatine, urea, lactate dehydrogenase (LDH), creatine kinase (CK), AST and ALT liver enzymes. Also, data about patients' follow-up (discharge, transfer from one hospital to another, death in less than 24 hours, death after 24 hours, treatment abandonment) were gathered. Biomarkers and other medical exams weren't requested for all patients, given that it was a choice made by the physician during the medical assistance on arrival. Prognostic biomarkers were calculated using blood tests at admission as follows: for Neutrophil-to-Lymphocyte Cell Ratio (NLCR), total neutrophils (total number/uL) were divided by total lymphocytes (total number/uL); for Platelet to Lymphocyte ratio (PLR), total platelet count (total number/uL) was divided by total lymphocytes; for Monocyte to Lymphocyte ratio (MLR), total monocytes were divided by total lymphocytes. Missing data was handled with only complete-case analysis. Then, a Pneumonia-Optimized Ratio (POR) score was created by selecting the biomarker with larger accuracy (NLCR) and multiplying it by the patients' CURB-65 score to improve its prognostic performance. All biomarkers were calculated at admission, and the primary outcome was death. Development of this score was performed according to TRIPOD guidelines [19].

Sample normality was evaluated with Shapiro-Wilk test. Sociodemographic data was described with median and interquartile range. Quantitative variables were compared with non-parametric tests, and qualitative variables were compared using chi-squared tests. Logistic regression was performed for each biomarker, and odds ratio was described individually and

adjusted accordingly. Receiver-operating characteristic (ROC) curve analysis was used to determine the optimal cut-off value of prognostic biomarkers of mortality. All analyses were performed with R 3.6.2 (R foundation for statistical computing, 2016).

## Results

There were a total of 646 individuals with complete neutrophil and lymphocyte data available in the inclusion criteria for this study. The median age of the cohort was 66 years (range, 18–103), 53.9% were female, and the median days of hospitalization was 6 (0–140). The percentage of individuals with CURB-65 scores was as follows: 0–1: 51.1%, 2: 29.58%, 3 or more: 19.3% (14 missing data, 2.16%). Characteristics of the sample are summarized in Table 1.

A total of 65 (10%) patients died during hospitalization in this analysis, of whom 7 (1%) in less than 24 hours of admission. Non-survivors showed increased days of hospitalization, CURB-65, rates of cancer and ischemic strokes (Table 1, p<0.05 for all). Prognostic biomarkers were calculated as described previously. The median scores for the entire cohort were as follows: NLCR 6.32 (0.06–137.6); PLR 180.47 (2.09–2385.7), MLR 57.59 (1.92–950). Non-survivors exhibited increased prognostic scores for all three biomarkers when compared with survivors (p<0.001, Table 1).

**Table 1. Baseline characteristics of the population studied.**

|  | Survivors | Non-survivors | p-values |
|---|---|---|---|
|  | (n = 581) | (n = 65) |  |
| Age (years) | 65 (18–100) | 78 (40–103) | **<0.001** |
| Sex (F) | 306 (52.7%) | 42 (64.6%) | 0.09 |
| Days of hospital | 6 (0–140) | 7 (0–119) | **0.03** |
| CURB-65 | 1.0 (0–5) | 2.0 (0–5) | **<0.001** |
| *Clinical comorbidities* |  |  |  |
| Hypertension | 68 (11.7%) | 5 (7.7%) | 0.45 |
| Diabetes | 41 (7%) | 3 (4.6%) | 0.73 |
| Chronic kidney disease | 66 (11.4%) | 2 (3.1%) | 0.06 |
| Heart failure | 85 (14.6%) | 9 (13.8%) | 1 |
| Infarction | 3 (0.5%) | 0 (0%) | 0.77 |
| Stroke | 52 (9.0%) | 18 (27.7%) | **<0.001** |
| Non-endstage cancer | 190 (32.5%) | 31 (47.7%) | **0.01** |
| Asthma | 46 (7.8%) | 3 (4.5%) | 0.56 |
| *Blood tests* |  |  |  |
| Hemoglobin (g/dL) | 12.1 (4.9–17.9) | 10.8 (3.2–17.5) | **<0.001** |
| White blood cells (x1000/uL) | 10.5 (0.39–74.38) | 11.8 (4.06–46.67) | **0.02** |
| Neutrophils (x1000/uL) | 7.7 (0.03–60.24) | 9.8 (2.8–45.64) | **<0.001** |
| Lymphocytes (x1000/uL) | 1.3 (0.10–59.67) | 0.9 (0.10–3.79) | **<0.001** |
| Blood urea (mg/dL) | 79.0 (1–139) | 85.0 (1–138) | 0.41 |
| Creatinine (mg/dL) | 0.8 (0.17–5.06) | 1.3 (0.41–5.83) | **<0.001** |
| *Prognostic biomarkers* |  |  |  |
| NLCR | 5.9 (0.06–98) | 12.8 (1.1–137.6) | **<0.001** |
| PLR | 177.0 (2.09–1716.98) | 233.6 (64.07–2385.76) | **<0.001** |
| MLR | 56.5 (1.92–800) | 89.5 (10–950) | **<0.001** |

Quantitative variables are shown in median (range).

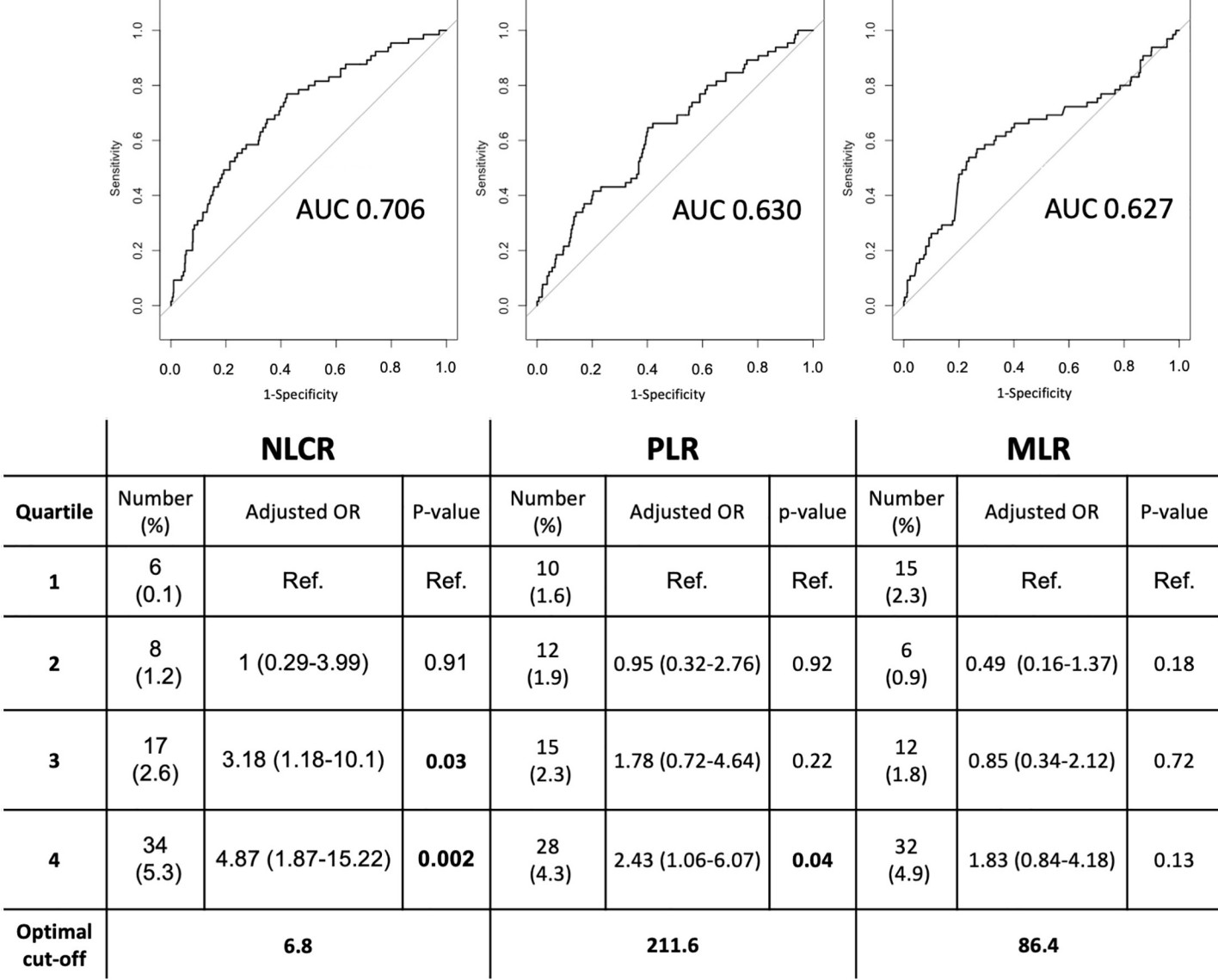

**Fig 1. Diagnostic properties of prognostic biomarkers divided in quartiles.** OR adjustments were made with age, serum creatinine, total white blood cells and hemoglobin. Ref.—Reference values. NLCR—Neutrophil-to-Lymphocyte Count Ratio. PLR—Platelet to Lymphocyte Ratio. MLR—Monocyte to Lymphocyte Ratio. AUC—Area under the curve.

In order to select the best prognostic biomarker of mortality in community-acquired pneumonia using the blood count variables, we have performed Receiver-Operating Curves (ROC) of each prognostic biomarker score aforementioned. Logistic regression of mortality in primary outcome is shown in Fig 1. There was a significant increase in crude mortality rate in the fourth quartile of NLCR and PLR (p = 0.005 unadjusted) after adjustment for age, serum creatinine, total white blood cells and hemoglobin (p = 0.01 adjusted). The fourth quartile of NLCR also showed an increased number of days of hospitalization when compared with the first quartile (11.06±15.96 vs. 7.02±8.39 respectively, p = 0.012).

An optimized biomarker was created multiplying NLCR by CURB-65 scores, which we coined POR (Pneumonia-Optimized Ratio). The optimal cutpoint for POR was 14.5 and the corresponding ROC for POR predicting mortality showed an AUC of 0.753. Sensibility of

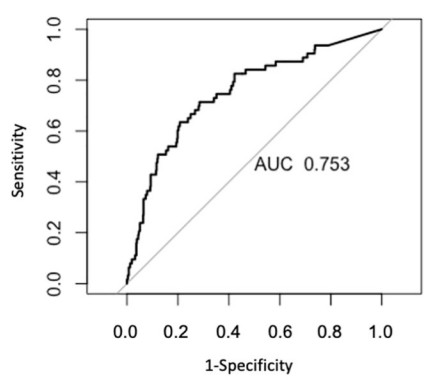

## Pneumonia-Optimized Ratio

| Quartile | Number (%) | Unadjusted OR (95% CI) | Adjusted OR (95% CI) | p-value |
|---|---|---|---|---|
| **1** | 6 (1) | Ref. | Ref. | Ref. |
| **2** | 4 (0.6) | 0.66 (0.16 - 2.34) | 0.49 (0.11 - 1.89) | 0.44 |
| **3** | 12 (1.9) | 2.17 (0.82 - 6.38) | 1.31 (0.41 - 4.45) | 0.35 |
| **4** | 40 (6.3) | 8.05 (3.53 - 21.70) | 3.3 (1.05 - 11.6) | **0.04** |

**Fig 2. Mortality rates in different quartiles of POR, unadjusted and adjusted models.** First quartile < 2.32; second Quartile: 2.32–7.42; third Quartile: 7.42–19.49; fourth Quartile > 19.49. Total of 630 individuals were included in the POR AUC analysis. Ref.—Reference values. OR—Odds Ratio. CI—Confidence Interval.

POR for mortality was 0.72 and specificity was 0.71. (Fig 2). The POR also showed increased crude mortality in the fourth quartile (POR>19.49) when compared with the first quartile (POR<2.32).

## Discussion

In this study, we have reassured the importance of prognostic biomarkers using typical blood tests in community-acquired pneumonia in a public hospital. Blood testing is widely available and minimally expensive in both outpatient and inpatient settings. Three different biomarkers were evaluated for prognostic accuracy in CAP, and we have created a novel one with increased prognostic performance in this study. Pneumonia-Optimized Ratio showed increased sensibility and specificity when compared with aforementioned biomarkers.

Neutrophil to lymphocyte cell ratio (NLCR) was the most accurate predictor of mortality in our sample when compared to previously described cell ratios (PLR, MLR). However, adding the CURB-65 scores to the NLCR increased the diagnostic accuracy into a novel biomarker that we coined Pneumonia-Optimized Ratio (POR). The latter improves the ability of clinicians to predict mortality with easily acquired patient's data, with absolutely no extra cost at admission. Patients identified at the admission with a higher NLCR or POR may raise awareness for early allocation of resources such as early reservation of an ICU bed or close monitoring of complications such as septic shock. Ultimately, the usage of POR may decrease mortality in patients presenting severe CAP. Importantly, low and middle-income countries present limited resources, and a prognostic biomarker with increased accuracy may easily be incorporated in the clinical practice and provide better outcomes [20].

Previous data have described that NLCR is an adequate prognostic marker of mortality from CAP in different settings. When compared with other prognostic at admission of a tertiary service, such as CRP and WBC, NLCR showed better prognostic performance after 30, 90 days [5] and overall [21]. In the intensive care setting, NLCR at the time of admission to the ICU was associated with 28-day mortality in patients without sepsis [15] and showed increased performance than other inflammation-based prognostic scores[17]. Other studies have demonstrated an association between lymphopenia and the sepsis syndrome [22] and an association between neutrophilia and mortality in patients with sepsis [23, 24]. In patients with bacteremia, a decreased eosinophil count and increased NLCR (above a median of 7) were associated with increased risk of mortality [25]. Unlike these studies that included only

patients admitted to a specific hospital unit, ours evaluated patients with CAP with different levels of severity, most not requiring ICU admission.

Regarding the public health system in low and middle-income countries, health depenses are decisive for equality and distribution of care. CAP is still a major public health issue in these countries, especially because of the costs of hospital admissions and severe cases [26]. The usage of simple prognostic biomarkers are stimulated due to their increased ability to predict death in such contexts [27]. The fourth quartile of the NLCR showed approximately 4 more days of hospitalization than the first quartile. This finding raises awareness of the increased financial burden of patients with higher NLCR, potentially investing higher resources in these individuals.

Despite our efforts, this study is not free from bias. Patients included in this study were from a tertiary public hospital, which may underestimate the applicability of described biomarkers in the outpatients setting, even though our sample showed different degrees of severity. Furthermore, individuals were retrieved in the hospital's database from the ICD-10 coding, which imposes the need of the assistant clinician to correctly describe all patients' comorbidities. Further studies may address the utility of these biomarkers in primary care.

## Conclusion

In sum, both Neutrophil to Lymphocyte Ratio and Pneumonia-Optimized Ratio are powerful prognostic biomarkers with great utility in clinical practice, especially in low- and middle-income countries with a public health system. The POR incorporated CURB-65 features and subsequently improved the prognostic accuracy for survival.

## Supporting information

**S1 Checklist. TRIPOD checklist: Prediction model development.**
(DOCX)

**S1 Fig. Prognostic properties of CURB-65 in our sample.** CURB-65 is an acronym for confusion, urea, respiratory rate, blood pressure and 65 years-old.
(DOCX)

## Author Contributions

**Conceptualization:** Renato Seligman.

**Data curation:** Vinícius Ferraz Cury, Lucas Quadros Antoniazzi, Paulo Henrique Kranz de Oliveira, Sainan Voss da Cunha, Guilherme Cristianetti Frison, Enrico Emerim Moretto.

**Formal analysis:** Wyllians Vendramini Borelli.

**Investigation:** Vinícius Ferraz Cury, Lucas Quadros Antoniazzi, Paulo Henrique Kranz de Oliveira, Sainan Voss da Cunha, Guilherme Cristianetti Frison, Enrico Emerim Moretto.

**Methodology:** Wyllians Vendramini Borelli, Renato Seligman.

**Supervision:** Renato Seligman.

**Writing – original draft:** Vinícius Ferraz Cury, Lucas Quadros Antoniazzi, Paulo Henrique Kranz de Oliveira, Wyllians Vendramini Borelli, Sainan Voss da Cunha, Guilherme Cristianetti Frison, Enrico Emerim Moretto.

**Writing – review & editing:** Vinícius Ferraz Cury, Lucas Quadros Antoniazzi, Paulo Henrique Kranz de Oliveira, Wyllians Vendramini Borelli, Sainan Voss da Cunha, Guilherme Cristianetti Frison, Enrico Emerim Moretto, Renato Seligman.

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
