## [Decision Letter · Decision Letter 0]

15 Feb 2021

PONE-D-21-02311

Developing the Pneumonia-Optimized Ratio for Community-acquired pneumonia: an easy, inexpensive and accurate prognostic biomarker

PLOS ONE

Dear Dr. Seligman,

Thank you for submitting your manuscript to PLOS ONE. After careful consideration, we feel that it has merit but does not fully meet PLOS ONE’s publication criteria as it currently stands. Therefore, we invite you to submit a revised version of the manuscript that addresses the points raised during the review process.

We look forward to receiving your revised manuscript.

Kind regards,

Aleksandar R. Zivkovic

Academic Editor

PLOS ONE

2. Thank you for providing the date(s) when patient medical information was initially recorded. Please also include the date(s) on which your research team accessed the databases/records to obtain the retrospective data used in your study.

3. We note that you have included a TRIPOD checklist, but have not included the page numbers where these items can be found. Please complete the checklist and upload it as a supplementary file.

Reviewers' comments:

Reviewer's Responses to Questions

**Comments to the Author**

1. Is the manuscript technically sound, and do the data support the conclusions?

Reviewer #1: Partly

Reviewer #2: Yes

2. Has the statistical analysis been performed appropriately and rigorously? 

Reviewer #1: No

Reviewer #2: Yes

3. Have the authors made all data underlying the findings in their manuscript fully available?

Reviewer #1: Yes

Reviewer #2: Yes

4. Is the manuscript presented in an intelligible fashion and written in standard English?

Reviewer #1: Yes

Reviewer #2: Yes

5. Review Comments to the Author

Reviewer #1: In the manuscript the authors present the results of a retrospective observational study. Receiver operating characteristic curve analysis was performed to show the utility of the biomarkers NLCR, MLR and PLR in predicting mortality in patients with CAP. The authors evaluated the correlation of these biomarkers level at admission with mortality and revealed whether the predictive power for mortality rates in patients with CAP could be improved by combining NLCR with CURB-65 scores. Based on NLCR multiplied by the CURB-65 score the authors created a new Pneumonia-Optimized Ratio (POR) score. POR based on a simple, cheap, and rapidly available biomarker NLCR does not require additional payment upon admission. This new POR score may be useful in identifying patients at high risk of mortality who require aggressive treatment.

General Comments:

1) How accurate was CURB-65 measured with AUC in terms of predicting mortality? What was the sensitivity and specificity of CURB-65 score for this cohort?

2) Authors include information about CURB-65 score for 97.8% patients: “CURB-65 score were low (score 0-1) for 50%, intermediate (score 2) for 28.9% and high (score 3-5) for 18.9%”. But it stays unclear what CURB-65 score had non-survivors patients. And more to the point it should be clear connection between this data and CURB-65 “range” in Table1 “1.0-2.0 For Survivors” and “2.0 - 3.0 for Non-survivors”

4) On the basis of survival over what period were the patients divided into 2 groups (Survival and Non-survival)?

5) It should be discussed how significant the difference is between AUC 0.706 and 0.753. How much better is the combined POR biomarker than the NLCR and CURB-65 alone?

6) The high percentage of patients with cancer (32.5% Survivors and 47.7% Non-survivors) should be discussed. Did the patients receive therapy, could this affect the results of the blood tests? Perhaps in table 1 it is worth indicating what types of cancers the patients had.

7) Why were mortality rates calculated for POR for 62 patients, while for biomarkers NLCR, PLR and MLR for 65 patients (see tables on Fig1 and Fig2, column “Number”)?

8) Authors should carefully check all the numerical values given in the work, as well as the correspondence of those indicated in the text and tables:

a. The Authors say that “The median age of the cohort was 66 years (IQR, 18-103), 53.9% were female, and the median days of hospitalization was 6 (0-140)” While in table 1 age range is 52.0-76.0 for Survivors and и 63.0-88.0 for Non-survivors. There are 18 and 103?

b. Also in table 1 the number of male “M” patients is 348 for Survivors and 42 for Non-survivors, which is inconsistent with 53.9% female patients in text.

c. Also the range for days of hospitalization in text (0-140) does not correspond to data in Table1 (ranges 2.0-10.0 and и 3.0-17.0).

d. Similar mismatch between the specified ranges in text and Table 1, for NLCR in Table1 (3.3-11.5 for Survivors), (7.1-22 for Non-survivors) and range (0.06-137.6) in text for the whole cohort. Range mismatch can also be seen for PLR and MLR.

e. In Abstract authors say that “Our sample consisted of 646 individuals” But next: “Patients scored 0-1 (323, 50%), 2 (187, 28.9%), or 3 or above (122, 18.9%) in the CURB-65, and 65 (10%) presented the primary outcome of death.” Totally it’s 632 patients.

f. All percentages in the "Number" column should be carefully checked (see tables in Fig. 1 and Fig. 2).

9) The abstract should contain "Conclusion", not "Discussion"

10) Also, the article would be more complete by adding the section "Conclusion"

Minor Corrections:

1) Throughout the text: pay attention to the spaces before and after the inserted citations

2) Perhaps for a better perception, it is worth presenting the X-axis in the generally accepted form of '1-Specificity' in Figures 1 and 2.

3) In Figure 1 “AUC: 0.753” the two-dot ":" should be removed so that Figure 1 and Figure 2 are signed uniformly

4) It would be better to make an explanation of "CKD" in table 1, and "Ref." in Figure 1 and Figure 2

5) In Table 1, it is better not to indicate tenths of the value for categories such as age and CURB-65, possibly also for hospitalization days.

Reviewer #2: There was a similar article published in Clin Lab. 2019 May 1;65(5). doi: 10.7754/Clin.Lab.2018.181042.

"Neutrophil-to-Lymphocyte Ratio in Adult Community-Acquired Pneumonia Patients Correlates with Unfavorable Clinical Outcome"

Ge YL et al. revealed NLR combined CURB-65 has better sensitivity and specificity (89.40% versus 91.30%) for unfavorable outcomes of CAP (ICU admission and 30-day mortality)

6. PLOS authors have the option to publish the peer review history of their article (what does this mean?). If published, this will include your full peer review and any attached files.

Reviewer #1: No

Reviewer #2: No

---

## [Author Response · Author response to Decision Letter 0]

5 Mar 2021

REVIEWER 1

Comment 1) "How accurate was CURB-65 measured with AUC in terms of predicting mortality? What was the sensitivity and specificity of CURB-65 score for this cohort?"

Answer. We would like to thank the reviewer for a careful reading and detailed contributions for our study.

We've measured CURB-65 accuracy, but we did not include in the manuscript because it's been widely addressed by other authors. However, we included our analysis in the supplementary material (Supplementary Figure 1). CURB-65 showed sensitivity and specificity for the outcome (death) of 0.76 and 0.54 respectively, and an AUC of 0.712.

Comment 2) "Authors include information about CURB-65 score for 97.8% patients: “CURB-65 score were low (score 0-1) for 50%, intermediate (score 2) for 28.9% and high (score 3-5) for 18.9%”. But it stays unclear what CURB-65 score had non-survivors patients. And more to the point it should be clear connection between this data and CURB-65 “range” in Table1 “1.0-2.0 For Survivors” and “2.0 - 3.0 for Non-survivors”"

Answer. Thanks for the opportunity in clarifying this information.

We've described the 95% range of each variable in Table 1. However, median CURB-65 scores for both Survivors and Non-Survivors are still the same presented in Table 1. 

We've previously calculated the CURB-65 scores for a total of 646 individuals of the sample. Actually, we've now corrected the percentage for the actual number of individuals with CURB (14 missing CURB-65, 2.16%).

We've corrected this information in the Results section, transcribed below (P. 6):

"The percentage of individuals with CURB-65 scores was as follows: 0-1: 51.1%, 2: 29.58%, 3 or more: 19.3% (14 missing data, 2.16%)."

We've also corrected Table 1 using the correct range. 

Comment 4) "On the basis of survival over what period were the patients divided into 2 groups (Survival and Non-survival)?"

Answer. The outcome was collected from patients admitted from 2019 to 2020. At the time of the analysis (October 2020), all patients had one of the following outcomes: Death (within 24 hours or not) or discharged (home or to a secondary care center). Thus, survival analysis included the whole hospitalization of all subjects.

Comment 5) It should be discussed how significant the difference is between AUC 0.706 and 0.753. How much better is the combined POR biomarker than the NLCR and CURB-65 alone?

Answer. Thank you for the opportunity to discuss this further. Indeed, an increased test accuracy allows a better identification of individuals with increased risk of death. Objectively, using the optimal cutpoint for both NLCR, CURB-65 and POR, we'd be able to identify, respectively, 59/100, 76/100 and 72/100. However, the rate of false positives is respectively 41.9%, 42%, and 28.47%. The difference between AUC may be modest, but it plays a massive impact in a public health context, such as the one described in this study.

We've discussed this result further in the Discussion section, 4th paragraph: Regarding the public health system in low and middle-income countries, health depenses are decisive for equality and distribution of care. CAP is still a major public health issue in these countries, especially because of the costs of hospital admissions and severe cases[25]. The usage of simple prognostic biomarkers are stimulated due to their increased ability to predict death in such contexts[26]. The fourth quartile of the NLCR showed approximately 4 more days of hospitalization than the first quartile. This finding raises awareness of the increased financial burden of patients with higher NLCR, potentially investing higher resources in these individuals.

Comment 6) The high percentage of patients with cancer (32.5% Survivors and 47.7% Non-survivors) should be discussed. Did the patients receive therapy, could this affect the results of the blood tests? Perhaps in table 1 it is worth indicating what types of cancers the patients had.

Answer. In the inclusion criteria, we defined to include only non-end stage cancer. We corrected in table 1.

Comment 7) Why were mortality rates calculated for POR for 62 patients, while for biomarkers NLCR, PLR and MLR for 65 patients (see tables on Fig1 and Fig2, column “Number”)?

Answer. The difference presented by different biomarkers is due to the missing CURB-65 data of 3 patients in this sample, which limited the application of POR accuracy analysis to 62 patients.

Comment 8) Authors should carefully check all the numerical values given in the work, as well as the correspondence of those indicated in the text and tables:

a. The Authors say that “The median age of the cohort was 66 years (IQR, 18-103), 53.9% were female, and the median days of hospitalization was 6 (0-140)” While in table 1 age range is 52.0-76.0 for Survivors and и 63.0-88.0 for Non-survivors. There are 18 and 103?

Answer. We want to thank the reviewer for this careful reading of the manuscript. Please see Answer to Comment 2 above. We've corrected the range in Table 1.

b. Also in table 1 the number of male “M” patients is 348 for Survivors and 42 for Non-survivors, which is inconsistent with 53.9% female patients in text.

Answer. Again, we apologize for the inconsistency between Table 1 and the body text. Table 1 refers to Female individuals (306+42 = 348. 348/646 = 53.9% in text).

We've corrected Table 1 accordingly.

c. Also the range for days of hospitalization in text (0-140) does not correspond to data in Table1 (ranges 2.0-10.0 and и 3.0-17.0).

Answer. We've corrected this information in Table 1. 

d. Similar mismatch between the specified ranges in text and Table 1, for NLCR in Table1 (3.3-11.5 for Survivors), (7.1-22 for Non-survivors) and range (0.06-137.6) in text for the whole cohort. Range mismatch can also be seen for PLR and MLR.

Answer. We've corrected this information in Table 1. 

e. In Abstract authors say that “Our sample consisted of 646 individuals” But next: “Patients scored 0-1 (323, 50%), 2 (187, 28.9%), or 3 or above (122, 18.9%) in the CURB-65, and 65 (10%) presented the primary outcome of death.” Totally it’s 632 patients.

Answer. Please see Answer Comment 2. We've corrected this information in the Results section.

f. All percentages in the "Number" column should be carefully checked (see tables in Fig. 1 and Fig. 2).

Answer. We've carefully checked all values described in Figure 1, Figure 2 and Table 1 as well. Small changes in the "Number" column regarding percentages (maximum change = 0.2%). 

We thank the reviewer for the opportunity to clarify and correct these values.

Comment 9) The abstract should contain "Conclusion", not "Discussion".

Answer. We've corrected this point in the abstract.

Comment 10) Also, the article would be more complete by adding the section "Conclusion"

Answer. We've corrected this point in the manuscript.

Comment "Minor Corrections":

1) Throughout the text: pay attention to the spaces before and after the inserted citations

2) Perhaps for a better perception, it is worth presenting the X-axis in the generally accepted form of '1-Specificity' in Figures 1 and 2.

3) In Figure 1 “AUC: 0.753” the two-dot ":" should be removed so that Figure 1 and Figure 2 are signed uniformly

4) It would be better to make an explanation of "CKD" in table 1, and "Ref." in Figure 1 and Figure 2

5) In Table 1, it is better not to indicate tenths of the value for categories such as age and CURB-65, possibly also for hospitalization days.

Answer. We've corrected these points in the manuscript. 

REVIEWER 2

Comment 1) "There was a similar article published in Clin Lab. 2019 May 1;65(5). doi: 10.7754/Clin.Lab.2018.181042.

"Neutrophil-to-Lymphocyte Ratio in Adult Community-Acquired Pneumonia Patients Correlates with Unfavorable Clinical Outcome"

Ge YL et al. revealed NLR combined CURB-65 has better sensitivity and specificity (89.40% versus 91.30%) for unfavorable outcomes of CAP (ICU admission and 30-day mortality)"

Answer. We would like to thank the reviewer for a detailed reading and analysis of the manuscript we submitted. 

Indeed, previous studies have mentioned biomarkers as predictors of mortality in Community-Acquired Pneumonia. Specifically the study conducted by Ge et al mentioned by the reviewer, our study distinguishes because of the population studied. We demonstrated that an NLR combined with CURB-65 is a useful predictor biomarker for mild cases and for patients with CAP hospitalized in the ICU.

We included a citation of this article in introduction, paragraph 2.

---

## [Editor Report · Decision Letter 1]

8 Mar 2021

Developing the Pneumonia-Optimized Ratio for Community-acquired pneumonia: an easy, inexpensive and accurate prognostic biomarker

PONE-D-21-02311R1

Dear Dr. Seligman,

We’re pleased to inform you that your manuscript has been judged scientifically suitable for publication and will be formally accepted for publication once it meets all outstanding technical requirements.

Kind regards,

Aleksandar R. Zivkovic

Academic Editor

PLOS ONE

---

## [Editor Report · Acceptance letter]

11 Mar 2021

PONE-D-21-02311R1 

Developing the Pneumonia-Optimized Ratio for Community-acquired pneumonia: an easy, inexpensive and accurate prognostic biomarker 

Dear Dr. Seligman:

I'm pleased to inform you that your manuscript has been deemed suitable for publication in PLOS ONE. Congratulations! Your manuscript is now with our production department. 

Kind regards, 

on behalf of

Dr. Aleksandar R. Zivkovic 

Academic Editor

PLOS ONE